# The diagnostic value of native kidney biopsy in low grade, subnephrotic, and nephrotic range proteinuria: A retrospective cohort study

Jonathan de Fallois[1]☯*, Soeren Schenk[1]☯, Jan Kowald[1], Tom H. Lindner[1], Marie Engesser[1], Johannes Münch[1,2], Christof Meigen[3], Jan Halbritter[1,2]

**1** Medical Department III, Division of Nephrology, University of Leipzig Medical Center, Leipzig, Germany, **2** Departement of Nephrology and Medical Intensive Care, Charité Universitätsmedizin Berlin, Berlin, Germany, **3** LIFE Child, Hospital for Children and Adolescents, Medical Faculty, Leipzig University, Leipzig, Germany

☯ These authors contributed equally to this work.
* Jonathan.defallois@medizin.uni-leipzig.de

**Data Availability Statement:** All relevant data are within the paper and its Supporting Information files.

## Abstract

### Background

In nephrotic range proteinuria of adult-onset, kidney biopsy is the diagnostic gold standard in determining the underlying cause of disease. However, in low grade or subnephrotic proteinuria the diagnostic value of kidney biopsy as first-line diagnostics is less well established.

### Methods

We conducted a retrospective analysis of all native kidney biopsies at our institution (n = 639) between 01/2012 and 05/2021 for comparison of histological diagnoses and clinical outcomes stratified by amount of proteinuria at the time of kidney biopsy: A: <300mg/g creatinine (low grade), B: 300-3500mg/g creatinine (subnephrotic), C >3500mg/g creatinine (nephrotic).

### Results

Nephrotic range proteinuria was associated with the highest frequency (49.3%) of primary glomerulopathies followed by subnephrotic (34.4%) and low grade proteinuria (37.7%). However, within the subnephrotic group, the amount of proteinuria at kidney biopsy was linearly associated with renal and overall survival (HR 1.05 per Δ100mg protein/g creatinine (95% CI: 1.02–1.09, p = 0.001)) independent of present histological diagnoses and erythrocyturia.

### Conclusion

Frequency of primary glomerulopathies supports to perform kidney biopsy in patients with subnephrotic proteinuria. These patients have a substantial risk of ESKD and death upon

**Funding:** The authors received no specific funding for this work.

**Competing interests:** The authors have declared that no competing interests exist.

follow-up. Therefore, diagnostic accuracy including histopathology is essential to guide personalized treatment and avert detrimental courses.

## Introduction

Quantifying proteinuria is a standard diagnostic tool in patients with chronic kidney disease (CKD) [1–6]. Protein excretion rates greater than 3.5 g per 24 hours or >3500 mg/g urine protein creatinine ratio (UPCR) are termed nephrotic range proteinuria and concomitant hypoalbuminemia, dyslipidemia, and edema define nephrotic syndrome (NS).

Adults with NS have a high risk for acute kidney injury (AKI) and progression to end-stage kidney disease (ESKD). High mortality is often attributed to cardiovascular disease [7–9]. Beyond NS, elevated urinary protein excretion in general represents an independent risk factor for increased cardiovascular disease and CKD-progression [10–14]. Interestingly, even subtle differences in very low grade albuminuria were recently shown to effect all-cause and cardiovascular mortality [15]. While the main underlying causes of primary and secondary NS are well characterized in adults [7, 16], in subnephrotic and low range proteinuria systematic biopsy data are scarce. This phenomenon is partly explained by the fact that the diagnostic value of native kidney biopsy is still less well-established in patients with subnephrotic range proteinuria (UPCR 300-3500mg/g crea (creatinine)) and little is known about the impact of kidney biopsy on differential diagnosis and prognosis. That is why most clinical studies focus on incidence and outcomes in nephrotic range proteinuria [7, 17, 18].

We conducted a retrospective single-center analysis of all native kidney biopsies at our medical institution over a nine-year time-period to help in comprehension of patients with subnephrotic or low grade proteinuria. We aimed at comparing frequencies of histological diagnoses stratified by proteinuria to subsequently evaluate clinical outcomes from electronic health records (EHR).

## Methods

### Study design and patients

In this retrospective observational study, we included all 639 patients, who underwent native kidney biopsy at the University of Leipzig Medical Center (Germany) from 01/2012 to 05/2021 (approval by local institutional review board (IRB) (250/21-ek)). Exclusion criteria were age < 18 years (n = 8) and absence of documented proteinuria (n = 24). At instances of repeated kidney biopsies (n = 34) only the first biopsy was included into the analysis. The remaining n = 573 patients with native kidney biopsies were stratified on the basis of UPCR in the following three groups: A: <300mg/g crea (low grade), B: 300-3500mg/g crea (subnephrotic), C: >3500mg/g crea (nephrotic) (**Fig 1**). We aimed at comparing frequencies of histological diagnoses and renal outcomes, stratified by amount of proteinuria at the time of kidney biopsy. We prespecified subgroup analysis referring to age, sex, preexisting hypertension and diabetes mellitus, obesity, percentage of glomerulosclerosis, proteinuria and erythrocyturia.

### Data collection

Study data were collected and managed using Research Electronic Data Capture (REDCap) tools hosted at Leipzig University [19]. The following clinical characteristics and demographic patient data were retrieved from EHR at the time of native kidney biopsy: age, sex, height, weight, BMI, preexisting history of diabetes, hypertension, kidney cysts, and positive family

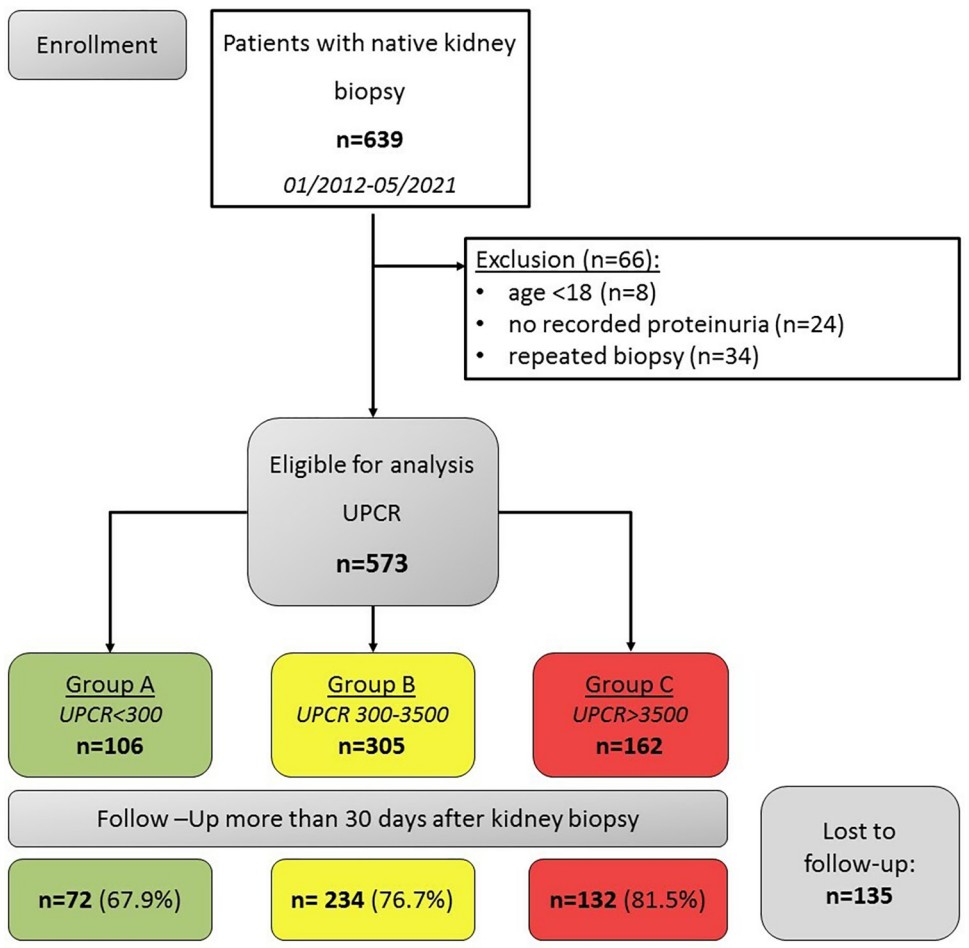

**Fig 1. Recruitment flow chart.** UPCR = urine protein creatinine ratio.

history for any kidney disease. From blood sampling the following laboratory parameters were collected: serum-creatinine, eGFR$_{creatinine}$ (CKD-EPI), urea, total serum protein, albumin, C3, C4, ANA, PR3 antibody, MPO antibody and Anti-glomerular basement membrane. Quantitative urinalysis was executed from spot urine samples prior to native kidney biopsy: UPCR, urine albumin creatinine ratio (UACR), alpha-1 microglobulin creatinine ratio, IgG creatinine ratio, glucose, urobilinogen, bilirubin, erythrocytes, leukocytes, and urinary casts.

We reviewed histological biopsy results and extracted diagnoses together with additional histological information. Available electron microscopy findings such as thin and irregular glomerular basement membrane, podocyte effacement and deposits were included. Clinical follow-up data were collected from the last documented hospital or outpatient contact (serum-creatinine, eGFR$_{creatinine}$, ESKD, defined as chronic requirement of kidney replacement therapy including, dialysis, kidney transplantation or eGFR$_{creatinine}$ <15ml/KG/1.72m$^2$ (ESKD-ND; end stage kidney disease–not on dialysis) for more than 28 days, and death). Safety endpoints related to biopsy-associated complications were documented.

## Statistical analysis

Metric variables were tested for normal distribution using the Shapiro–Wilk test. Non normally distributed variables are given as median with 25th and 75th quantile in square brackets

and continuous variables as mean with standard deviation. Categorical variables are displayed as frequencies and percentages. Normally distributed variables were tested by a one-way ANOVA and Bonferroni correction. Categorical variables were tested by Kruskal-Wallis test. Kaplan–Meier survival curve was used to calculate and depict the survival function of patient outcome. Cox proportional hazard models were used to adapt for potential effects of modifiers relating to the composite outcome. In model 1 we used an univariate model. Model 2 considered biometric factors (age, gender, body mass index (BMI)) and model 3 preexisting conditions (diabetes, hypertension, baseline eGFR). Influence on composite endpoint by different variables were tested by binary regression analysis. A cubic spline curve was calculated to reflect the connection between hazard ratios and UPCR values. The significance level was defined as 5%. Statistical analysis was performed using IBM SPSS, version 26 (Minneapolis, USA). GraphPadPrism, version 9 (San Diego, CA) and R software, version 3.6.3 (www.r-projectorg; R Foundation for Statistical Computing, Vienna) with "survival" package (Version 3.2) were applied for the generation of graphs.

## Results

### Patient characteristics at baseline

According to their UPCR at kidney biopsy, 106 patients were assigned to group A (low grade), 305 to group B (subnephrotic), and 162 to group C (nephrotic). Baseline characteristics such as sex, age, BMI, diabetes mellitus, and hypertension were comparable (non-significant) between groups (**Table 1**).

However, group B and C displayed higher baseline serum-creatinine values (lower eGFRs) compared to group A. Serum albumin was lowest in group C and highest in group A, suggesting pronounced urinary protein loss in subnephrotic and nephrotic groups. Erythrocyturia was more frequent among patients in group B and C compared to group A. Differentiation of proteinuria showed additional considerable higher urine alpha-1 microglobulin creatinine ratio and IgG creatinine ratio levels in group B and C besides the preset higher UACR levels.

Additional patient characteristics at baseline are displayed in **Table 1**.

### Nephropathy spectrum across proteinuric subgroups

The frequency of primary glomerulopathies (membranous nephropathy (MN) > immunoglobulin A nephropathy (IgAN) > minimal change disease (MCD) > focal segmental glomeruloclerosis (FSGS) > Alport syndrome/thin basement nephropathy (TBMN)) correlated with the amount of proteinuria. Hence, primary glomerulopathies were found most frequently in nephrotic 80/162 (49.3%), followed by low grade 40/106 (37.8%) and subnephrotic group 105/305 (34.4%).

In contrast, secondary nephropathies such as interstitial nephritis, Lupus nephritis, ANCA associated glomerulonephritis, thrombotic microangiopathy, acute tubular necrosis, amyloidosis, plasma cell dyscrasia and others, excluding hypertensive and diabetic glomerulopathy were commonest in group A 58/106 (54.7%); closely followed by group B 158/305 51.8% and group C 50/162 (30.9%). Specifically, systemic ANCA-associated vasculitis and lupus nephritis were mostly part of the subnephrotic group: 65/305 (21.3%), but only present in 6.6% and 10.5% of groups A and C respectively.

Remarkable diabetic nephropathy had by far the highest frequency among group C 23/162 (14.1%) compared to 5/106 (4.7%) and 15/305 (4.9%) within group A and B. Frequency of hypertensive nephropathy was more likely within the groups with higher proteinuria.

Detailed distribution of histological diagnoses is illustrated in **Fig 2** and **S1 Table**.

**Table 1. Patient baseline characteristics at native kidney biopsy.**

| Patient characteristics | Group A | Group B | Group C | |
|---|---|---|---|---|
| | UPCR <300mg/g creatinine | UPCR 300–3500 mg/g creatinine | UPCR >3500 mg/g creatinine | p |
| | n = 107 | n = 306 | n = 163 | |
| Age (y) | 53.26±16.42 | 53.43±17.46 | 54.00±17.60 | 0.873 |
| Sex (male) | 48 (45.3%) | 131 (43.0%) | 68 (42.0%) | |
| Height (cm) | 172.60±9.07 | 172.23±9.47 | 172.05±8.86 | 0.738 |
| Actual body weight (kg) | 80.41±15.12 | 80.79±19.86 | 82.51±22.03 | 0.727 |
| BMI (kg/m2) | 26.96±4.59 | 27.21±6.33 | 27.89±7.51 | 0.589 |
| *Medical history*: | | | | |
| Diabetes mellitus | 20 (18.7) | 69 (22.5) | 46 (28.2) | 0.169 |
| Hypertension | 63 (58.9) | 211 (69) | 117 (71.8) | 0.072 |
| Family history of kidney disease | 15 (14) | 7 (2.3) | 2 (1.2) | <0.005 |
| Kidney cysts | 4 (3.7) | 28 (9.2) | 15 (9.2) | 0.181 |
| *Laboratory parameters*: | | | | |
| Serum creatinine (µmol/l) | 152 [100; 255] | 212 [117; 382] | 194 [118; 343] | 0.001 |
| eGFR$_{creatinine}$ (CKD-EPI) ml/kg/1.72m$^2$ | 40 [21; 65] | 25 [13; 59] | 30 [15; 57] | 0.03 |
| Urea | 10 [7; 16] | 14 [9;19] | 12 [9; 18] | 0.07 |
| Serum albumin (g/l) | 42 [37; 44] | 37 [30; 42] | 30 [23; 34] | <0.005 |
| UPCR (mg/g creatinine) | 149 [83; 217] | 1213 [627; 1888] | 6000 [4720; 6734] | <0.005 |
| UAPR (mg/g creatinine) | 46 [20; 117] | 799 [396; 1471] | 4400 [3606; 5056] | <0.005 |
| Urine α1-MG/g creatinine | 21 [1; 53] | 71 [25;136] | 88 [53;158] | <0.005 |
| Urine IgG/g creatinine | 7 [0; 14] | 78 [36;157] | 430 [239; 961] | <0.005 |
| Urine erythrocytes/µl | 14 [3; 66] | 41 [7; 187] | 20 [5; 151] | 0.006 |
| Urine leukocytes/µl | 9 [3; 26] | 15 [6;89] | 11 [5; 70] | 0.002 |
| Erythrocyturia | 42 (39.6) | 174 (57.0) | 77 (47.5) | 0.005 |

Data presented as n (%), mean ± standard deviation, median [25$^{th}$; 75$^{th}$ quantile] *y* years, *BMI* body mass index, *eGFR (CKD-EPI)* estimated glomerular filtration rate (Chronic Kidney Disease Epidemiology Collaboration), *UPCR* urine protein creatinine ratio, *UACR* urine albumin creatinine ratio, *α1-MG* alpha 1 microglobulin, IgG Immunoglobulin G.

## Clinical outcome

Biopsy related complications were similarly rare in all three groups (S2 Table). Data on the composite clinical endpoint consisting of all-cause mortality and/or ESKD and/or >50% decrease from baseline eGFR after more than 30 days post-kidney biopsy were available in 72 (67.9%) patients in group A; 234 (76.7%) in group B, and 132 (81.5%) in group C, respectively (Fig 1). The average period to the last available clinical outcome data was 755 days (CI: 585–925) in group A, 694 days (CI: 596–793) in group B and 728 days (CI: 583–874) in group C (p = 0.594). By Kaplan-Meier survival analysis, the greatest likelihood of reaching the composite endpoint was found in group C. However, also patients with subnephrotic proteinuria were found to show significantly higher rates of composite endpoints compared to group A (log-rank = 0.001). Thus, risk of all-cause mortality, >50% decrease from baseline eGFR, and progression to ESKD were both higher in the subnephrotic group compared to the group with low grade proteinuria (Fig 3A, Table 2).

Rigorous analysis of single outcome parameters showed that all-cause mortality was significantly higher in the nephrotic group compared to both other groups. Progression to ESKD and >50% decrease from baseline eGFR were markedly increased in both the nephrotic and the subnephrotic group. However, comparison of ESKD frequency between subnephrotic and nephrotic patients did not differ significantly (Fig 3B–3D).

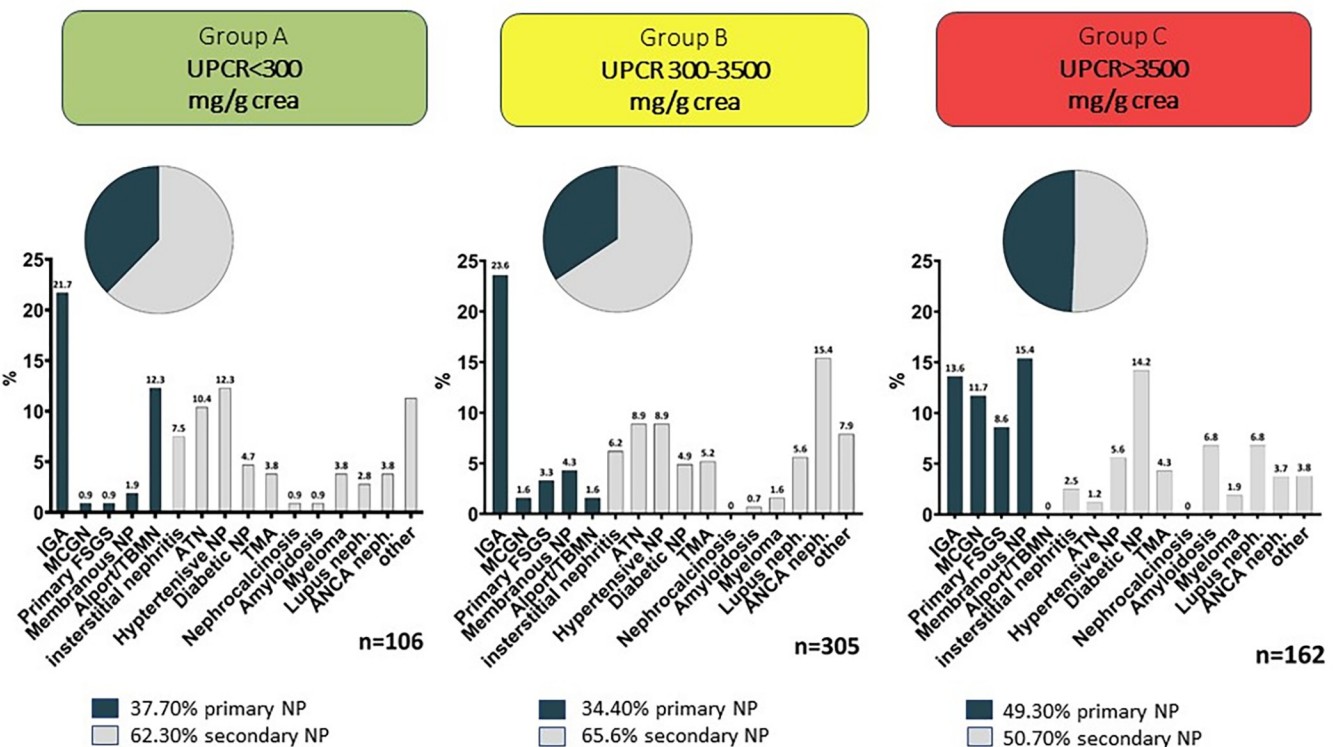

**Fig 2. Frequencies of diagnoses.** UPCR = urine protein creatinine ratio; IgAN = Immunoglobulin A Nephritis; MCGN = minimal change glomerulonephritis; NP = nephropathy; TBMN = thin basement nephropathy; ATN = acute tubular necrosis; TMA = thrombotic microangiopathy; ANCA = anti-neutrophil cytoplasmic antibody.

Accounting for about 40% of all cases, infectious complications were the leading cause of death in all study groups; followed by cardiovascular disease, especially in the nephrotic group (**S3 Table**).

In order to adjust for potential confounders, we performed multivariate Cox proportional regression models comparing nephrotic versus subnephrotic proteinuria. Results demonstrated that UPCR is a reliable and independent predictor of kidney outcome, with regard to model 1 to 3 (**Table 3**).

## Subgroup analyses

Exploration of suspected key determinants on composite endpoints showed that preexisting health conditions such as diabetes mellitus influenced patient outcome in both groups. Arterial hypertension and age affected outcomes in group B, but not in group C. Amount of proteinuria determined outcomes in group B (HR 95%: 1.05, CI: 1.02–1.09; p = 0.001) compared to group C (HR 95%: 1.01, CI: 0.99–1.01; p = 0.164). Presence of erythrocyturia at kidney biopsy did not influence the composite endpoints (HR 95%: 1.02, CI: 0.94–1.11; p = 0.596 within group B and HR 95%: 1.10, CI: 0.92–1.31; p = 0.287 within group C).

Other determinants are displayed in Table 4.

## Proteinuria related hazard ratios

Illustrating the significant increasing risk for the composite outcome, ESKD and all-cause mortality, adjusted hazard ratios were calculated with regard to UPCR. The results are plotted

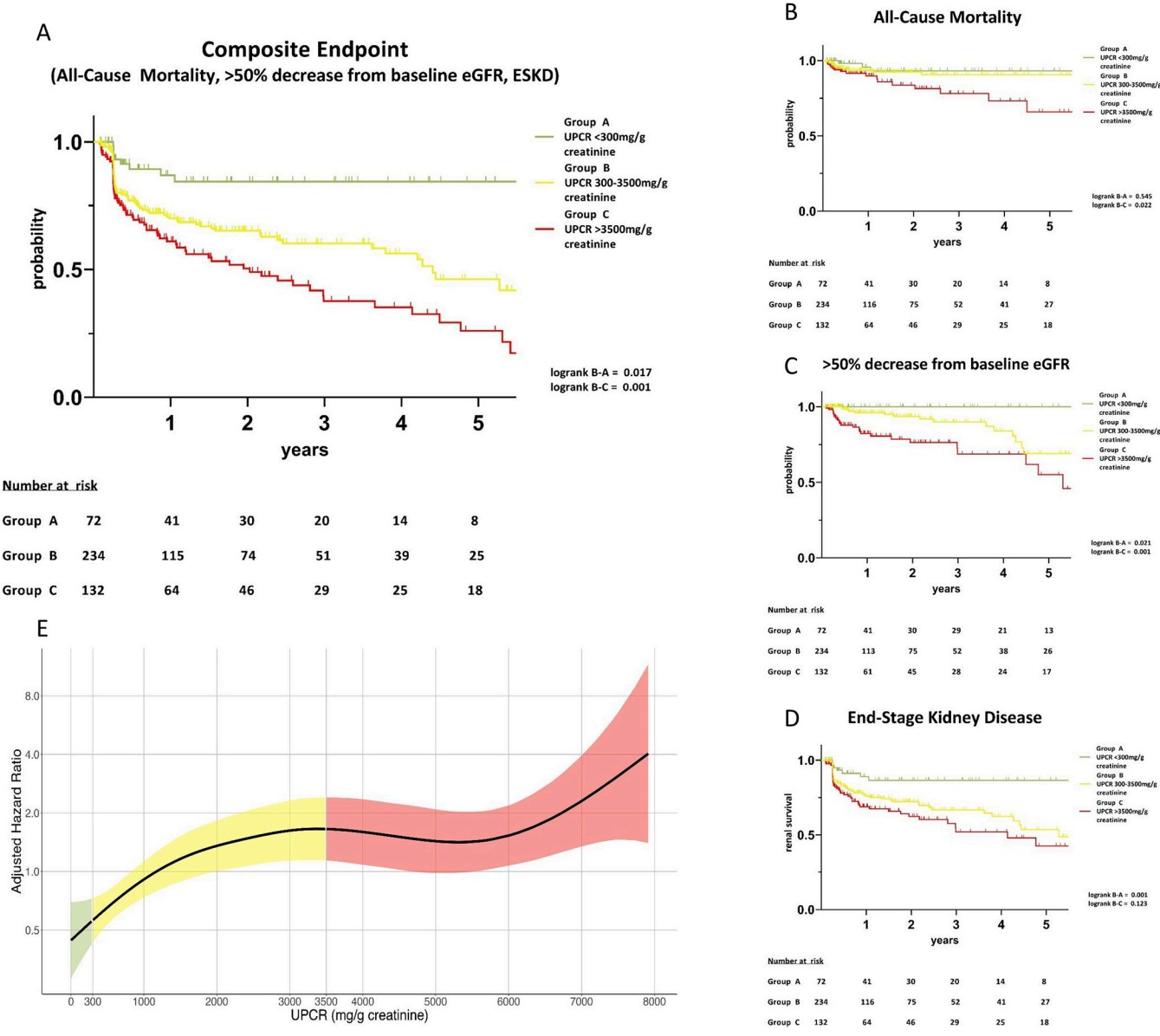

**Fig 3.** a Kaplan-Meier survival analysis regarding composite endpoint (all-cause mortality, decrease of eGFR by >50% and ESKD) ESKD = end stage kidney disease. b Kaplan-Meier survival analysis regarding all-cause mortality end-stage kidney disease. c Kaplan-Meier survival analysis regarding >50% decrease from baseline eGFR. d Kaplan-Meier survival analysis regarding ESKD. e Cubic spline curve analysis reflecting adjusted hazard ratios and UPCR UPCR urine protein creatinine ratio.

by a cubic spline curve analysis (**Fig 3E**). Interestingly, adjusted hazard ratios rose in an almost linear fashion within the subnephrotic range and showed no further increase from about 3500–6000 mg/g crea. UPCR above 6000 mg/g crea though was associated with additional risk of ESKD and death (**Fig 3E**).

# Discussion

The present analysis demonstrates a striking and in part linear association of quantitative proteinuria at the time of kidney biopsy with subsequent renal and patient survival (**Fig 3A–3D**).

Table 2. Patient outcomes by proteinuria-defined subgroup.

| Outcome parameter | Group A <300mg/g creatinine n = 72 (67.9%) | Group B 300-3500mg/g creatinine n = 234 (76.7%) | Group C >3500mg/g creatinine n = 132 (81.5%) |
|---|---|---|---|
| Composite endpoint (all-cause mortality, ESKD or >50% eGFR decrease) | 9 (12.5%) | 85 (36.3%) | 68 (51.5%) |
| All-cause mortality | 5 (6.9%) | 18 (7.7%) | 21 (15.9%) |
| >50% decrease from baseline eGFR | 0 | 17 (7.3%) | 28 (21.2%) |
| ESKD | 4 (5.6%) | 65 (27.8%) | 47 (35.6%) |
| • ESKD-ND | 0 | 13 | 8 |
| • Chronic intermittent dialysis | 3 | 49 | 32 |
| • Kidney transplantation | 1 | 3 | 7 |

Data shown: n (%), ESKD: end stage kidney disease, eGFR: estimated glomerular filtration rate, ESKD-ND: end stage kidney disease–not on dialysis (eGFR$_{creatinine}$ <15ml/min/1.73m$^2$).

Although subnephrotic proteinuria is less established as kidney biopsy indication, we found about one third of these patients being diagnosed with a primary kidney disease upon histological analysis (Fig 2). Moreover, subnephrotic proteinuria was associated with a significantly increased risk of progression to ESKD compared to patients with low grade proteinuria.

To date, there are only few clinical studies focusing on patients with subnephrotic proteinuria upon biopsy proven nephropathies. In line with our study, recently published biopsy cohorts comprised about 50% of cases with subnephrotic proteinuria (53% versus 44.77% and 47.6%) [20, 21]. Histological results in this studies showed that IgAN is most frequent (20.9% and 35.4%) in the group of subnephrotic patients similar to our results [20, 21]. In detail, the frequency of following diagnosis differed from our study. FSGS and lupus nephritis were common diagnoses within the cohort from Kuwait [20], whereas within the cohort from India only lupus nephritis was frequent [21]. In contrast to these studies we observed considerable more patients with ANCA associated glomerulonephritis [20, 21]. These findings are probably related to the different genetic backgrounds of the patient populations. For example, it is a well-established fact that incidence and prevalence of systemic lupus erythematosus varies geographically [22].

As expected, patients with nephrotic range proteinuria showed the highest mortality rates and the highest risk of CKD-progression independent of the underlying histological diagnosis. In patients with NS, thromboembolism and cardiovascular morbidity is the main cause of premature death [7–9, 17, 23], likely based on hypercoagulability with urinary loss of endogenous anticoagulants, increased synthesis of procoagulants, and endothelial dysfunction [24–26].

Table 3. Risk of all-cause mortality and progression to end-stage kidney disease.

| | Group B | Group C | |
|---|---|---|---|
| Combined endpoint | HR (95%CI) | HR (95%CI) | p |
| Model 1 | 1.0 (reference) | 1.50 (1.07–2.11) | 0.018 |
| Model 2 | 1.0 (reference) | 1.51 (1.08–2.14) | 0.017 |
| Model 3 | 1.0 (reference) | 1.59 (1.12–2.27) | 0.010 |
| Number of events (%) | 83 (32.2) | 68 (47.9) | |

HR: Hazard ratio, CI: Confidence interval, Combined endpoint: All-cause mortality + end stage kidney disease.
Model 1: Unadjusted / Model 2: Adjusted for age, sex, BMI / Model 3: Model 2 + diabetes, hypertension, baseline eGFR.

**Table 4. Subgroup analysis of subnephrotic range proteinuria.**

| Subgroup | Group B | | Group C | |
|---|---|---|---|---|
| | HR (95% CI) | p | HR (95% CI) | p |
| Sex (male) | 0.69 (0.40–1.18) | 0.173 | 0.48 (0.25–9.54) | 0.036 |
| Hypertension | 1.85 (1.02–3.35) | 0.044 | 1.59 (0.75–3.39) | 0.227 |
| Diabetes | 2.12 (1.16–3.88) | 0.015 | 2.34 (1.09–5.03) | 0.030 |
| Age >55 years | 2.02 (1.17–3.46) | 0.011 | 1.31 (0.67–2.57) | 0.432 |
| Obesity BMI:>30 kg/m$^2$ | 1.39 (0.82–2.36) | 0.226 | 0.67 (0.31–1.43) | 0.301 |
| Glomerulosclerosis | 1.03 (1.02–1.04) | 0.001 | 1.05 (1.03–1.07) | 0.001 |
| Proteinuria | 1.05 (1.02–1.09) | 0.001 | 1.01 (0.99–1.01) | 0.164 |
| Erythrocyturia | 1.02 (0.94–1.11) | 0.596 | 1.10 (0.92–1.31) | 0.287 |

HR Hazard ratio; CI Confidence interval; glomerulosclerosis: percent of sclerosed glomeruli; proteinuria per Δ100mg protein/g creatinine BMI: body mass index.

Moreover altered expression and activity of lipoproteins lead to an increased risk of atherosclerosis [27]. Lastly, humoral immunodeficiency through urinary immunoglobulin wasting and immunological dysfunction due to immunosuppressant intake raise the risk for fatal infection [28, 29]. Indeed, in our study fatal infections with about 40% were the leading cause for dead in the study population and cardiovascular mortality was higher in the nephrotic group (**S3 Table**).

The liver tries to balance protein loss in NS by increasing protein and albumin synthesis to keep the homeostasis [30]. Urinary protein loss in subnephrotic patients exceeds not the compensation thresholds of the liver, but is still associated with high morbidity and progression to ESKD. According to the standard plasma albumin values in most patients with subnephrotic proteinuria other clinically relevant dysregulated metabolic processes involving blood coagulation, endothelial function, lipid metabolism, and immune system are to be suspected. Recent registry data showed that even diabetes patients with microalbuminuria and without cardiovascular disease bear a significantly increased risk for ischemic stroke, myocardial infarction, and all-cause mortality compared to patients without albuminuria (hazard ratio (HR): 1.28 (95% CI: 1.07–1.52) [31]. Furthermore, studies in healthy individuals demonstrated that changes of albuminuria are associated with higher risks of heart failure and all-cause mortality especially within low grade proteinuria at normal range [32, 33]. Reduction of albuminuria below 300 mg/g creatinine was shown to reduce all-cause mortality in patients with diabetes (HR: 0.69; 95% CI 0.52–0.91; p = 0.008) [34]. Notably, reduced UPCR in subnephrotic range proteinuria was associated with nearly linear reduction of hazard ratios for ESKD and mortality in this study (**Fig 3E**). It stresses feasible benefits of antiproteinuric therapeutic interventions within this group.

Our subgroup analysis indicated that preexisting arterial hypertension has an effect on kidney and patient outcomes especially in the subnephrotic group, underlining the importance of blood pressure control.

Interestingly, all-cause mortality was significantly higher in the nephrotic group compared to the subnephrotic group, while progression to ESKD was similarly frequent in both groups. This may be due to the fact that acute thromboembolic events with fatal outcome were more likely to occur in patients with NS. However, absolute differences in all-cause mortality must be interpreted with caution for the limited study population and a relatively short follow-up period in our study (**Fig 3A–3D**).

Besides nephrotic proteinuria, KDIGO recommends kidney biopsy in glomerular erythrocyturia [1]. In this study erythrocyturia was most frequent within the subnephrotic group.

However, subgroup analysis revealed that the additional presence of erythrocyturia was not associated with progression to ESKD and all-cause mortality (**Table 4**). Risk prediction regarding clinical outcomes seems complicated, because quantifying erythrocyturia systematically is more difficult. The observed biopsy associated complication rate did not differ between groups and was found in the range of previously reported biopsy studies (**S2 Table**) [35, 36].

Also, the rate of mortality and progression to ESKD in patients with NS was similar to other studies (our 42% vs 40.7%) [17], reflecting representative composition of the study population. However, this study has several limitations. First, it is a monocentric and retrospective analysis and based on this, the indication to proceed to kidney biopsy was made by different attending nephrologists on a non-standardized but individual professional basis. Second, in most cases only results of spot urine samples were available with the caveat of moderate correlation to protein excretion in a 24h urine sample [1, 2, 4, 5]. Third, the study is limited by its moderate cohort size not allowing for histology based subgroup comparisons. Additionally, the relatively short observational period of about 700 days on average limits statements on long-term patient outcome and potential attenuation by therapeutic efforts. Clinical endpoints in mildly and moderately progressive disease would have only been captured upon long-term observation.

## Conclusion

Adult patients with subnephrotic proteinuria have a similarly severe risk of progression to ESKD when compared with nephrotic range proteinuria. Therefore, kidney biopsy constitutes an indispensable diagnostic tool to ascertain the underlying glomerulopathy independent of default proteinuria cut-off values. Histological work-up together with molecular genetics enable establishment of precise diagnoses and pave the way to appropriate therapeutic management to avert fast progressive loss of renal function.

## Supporting information

**S1 Table. Biopsy diagnosis.**
(DOCX)

**S2 Table. Biopsy complications.**
(DOCX)

**S3 Table. Specific causes of mortality.**
(DOCX)

## Author Contributions

**Conceptualization:** Jonathan de Fallois, Soeren Schenk, Johannes Münch, Jan Halbritter.

**Data curation:** Jonathan de Fallois, Soeren Schenk, Jan Kowald, Marie Engesser.

**Formal analysis:** Soeren Schenk.

**Investigation:** Jonathan de Fallois, Soeren Schenk, Jan Kowald.

**Methodology:** Jonathan de Fallois, Soeren Schenk, Jan Kowald, Jan Halbritter.

**Project administration:** Jonathan de Fallois, Soeren Schenk, Christof Meigen, Jan Halbritter.

**Resources:** Jonathan de Fallois, Jan Kowald, Tom H. Lindner, Jan Halbritter.

**Software:** Jonathan de Fallois, Soeren Schenk, Christof Meigen.

**Supervision:** Jonathan de Fallois, Johannes Münch, Jan Halbritter.

**Validation:** Jonathan de Fallois, Tom H. Lindner, Jan Halbritter.

**Visualization:** Jonathan de Fallois, Soeren Schenk, Christof Meigen.

**Writing – original draft:** Soeren Schenk.

**Writing – review & editing:** Jonathan de Fallois, Jan Kowald, Tom H. Lindner, Marie Engesser, Johannes Münch, Christof Meigen, Jan Halbritter.

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
