## [Decision Letter · Decision Letter 0]

23 Jun 2022

PONE-D-22-13404The diagnostic value of native kidney biopsy in low grade, subnephrotic, and nephrotic range proteinuria: a retrospective cohort studyPLOS ONE

Dear Dr. de Fallois

Thank you for submitting your manuscript to PLOS ONE. After careful consideration, we feel that it has merit but does not fully meet PLOS ONE’s publication criteria as it currently stands. Therefore, we invite you to submit a revised version of the manuscript that addresses the points raised during the review process.

We look forward to receiving your revised manuscript.

Kind regards,

Xianwu Cheng, M.D., Ph.D., FAHA

Academic Editor

PLOS ONE

Journal Requirements:

Additional Editor Comments:

Both reviewers have concerned the statistical analysis.

Reviewers' comments:

Reviewer's Responses to Questions

**Comments to the Author**

1. Is the manuscript technically sound, and do the data support the conclusions?

Reviewer #1: Yes

Reviewer #2: Yes

2. Has the statistical analysis been performed appropriately and rigorously? 

Reviewer #1: Yes

Reviewer #2: Yes

3. Have the authors made all data underlying the findings in their manuscript fully available?

Reviewer #1: Yes

Reviewer #2: Yes

4. Is the manuscript presented in an intelligible fashion and written in standard English?

Reviewer #1: Yes

Reviewer #2: Yes

5. Review Comments to the Author

Reviewer #1: The authors conducted a retrospective analysis of all native kidney biopsies for comparison of histological diagnoses and clinical outcomes stratified by amount of proteinuria at the time of kidney biopsy.

They concluded that frequency of primary glomerulopathies supports to perform kidney biopsy in patients with subnephrotic proteinuria.

The study is interesting and important. The results of this study are convincing. The manuscript is well written and easy to follow.

Concerns:

1) In the abstract, results section, one of 37.7 should be 34.4.

2) The authors should indicate what the results of the genetic analysis were in this study cohort and how they relate to the data already presented in this study.

3) As a limitation of this study, the authors point out that the observation period is short, and they should also discuss what impact this may have.

Reviewer #2: In this retrospective study authors tried to evaluate the utility of kidney biopsy in patients with low grade or subnephrotic proteinuria in comparison to the established value of biopsy in patients with nephrotic range proteinuria.

The results presented in this manuscript are based on original research and the whole article is well written and intelligible. The statistics and the overall design of the study are appropriate and are described in sufficient detail. The conclusions are presented appropriately and are supported by the analysis.

Nevertheless, in table 1 authors must show the p value for any statistically significant differences between the presented variables.

Authors may opt to present patients with eGFR < 15 ml/min/1.73 m2 as CKD-5ND (not on dialysis).

Authors describe increased all cause mortality for patients with nephrotic range proteinuria. Are there any data on specific causes of mortality (i.e. cardiovascular disease, cancer etc.)? Can authors describe which are the specific causes of this increased mortality?

As the observational period is limited, can authors include in their analyses the decrease of eGFR by 50% as an end point? This way it will be possible to detect the probability of kidney function deterioration even in patients with low grade proteinuria.

6. PLOS authors have the option to publish the peer review history of their article (what does this mean?). If published, this will include your full peer review and any attached files.

Reviewer #1: No

Reviewer #2: **Yes: **Marios Papasotiriou MD, PhD

---

## [Author Response · Author response to Decision Letter 0]

6 Jul 2022

July 2022

POINT-TO-POINT RESPONSE – Manuscript PONE-D-22-13404

Review Comments to the Author

Reviewer #1: 

The authors conducted a retrospective analysis of all native kidney biopsies for comparison of histological diagnoses and clinical outcomes stratified by amount of proteinuria at the time of kidney biopsy. They concluded that frequency of primary glomerulopathies supports to perform kidney biopsy in patients with subnephrotic proteinuria. The study is interesting and important. The results of this study are convincing. The manuscript is well written and easy to follow.

RESPONSE: We thank the Reviewer for the appreciation.

Concerns:

1) In the abstract, results section, one of 37.7 should be 34.4.

RESPONSE: Thank you for pointing that out. We corrected accordingly.

2) The authors should indicate what the results of the genetic analysis were in this study cohort and how they relate to the data already presented in this study.

RESPONSE: Although genetic testing was performed in 45/573 (7.9%) of the total cohort, genetic analysis was not systematically performed as part of the study protocol. Therefore, obtained genetic diagnoses were not reported and statistically associated with kidney biopsy results and degree of proteinuria. However, this comparison would be an interesting follow-up investigation beyond the current manuscript.

3) As a limitation of this study, the authors point out that the observation period is short, and they should also discuss what impact this may have.

RESPONSE: We fully agree and added the following paragraph to the Discussion: 

“Additionally, the relatively short observational period of about 700 days on average limits statements on long-term patient outcome and potential attenuation by therapeutic efforts. Clinical endpoints in mildly and moderately progressive disease would have only been captured upon long-term observation.“

Reviewer #2: 

In this retrospective study authors tried to evaluate the utility of kidney biopsy in patients with low grade or subnephrotic proteinuria in comparison to the established value of biopsy in patients with nephrotic range proteinuria. The results presented in this manuscript are based on original research and the whole article is well written and intelligible. The statistics and the overall design of the study are appropriate and are described in sufficient detail. The conclusions are presented appropriately and are supported by the analysis. 

RESPONSE: We thank the Reviewer for the appreciation.

Nevertheless, in table 1 authors must show the p value for any statistically significant differences between the presented variables.

RESPONSE: As suggested, we added the p-values in Table 1.

Authors may opt to present patients with eGFR < 15 ml/min/1.73 m2 as CKD-5ND (not on dialysis).

RESPONSE: We agree and adapted the proposed terminology in Tab. 2 and throughout the manuscript.

Authors describe increased all-cause mortality for patients with nephrotic range proteinuria. Are there any data on specific causes of mortality (i.e. cardiovascular disease, cancer etc.)? Can authors describe which are the specific causes of this increased mortality?

RESPONSE: We now provide supplementary information about specific causes of mortality in a new supplementary Table (Suppl. Table 3). In all study groups infectious complications predominated. Interestingly, cardiovascular disease was most frequent in the nephrotic group. We added the following sentence to the Results section:

“Accounting for about 40% of all cases, infectious complications were the leading specific cause of death in all study groups; followed by cardiovascular disease, especially in the nephrotic group (Suppl. Tab. 3).”

As the observational period is limited, can authors include in their analyses the decrease of eGFR by 50% as an end point? This way it will be possible to detect the probability of kidney function deterioration even in patients with low grade proteinuria.

RESPONSE: We thank the Reviewer for this valuable suggestion. Accordingly, we now calculated a new composite outcome including decrease of >50% from baseline eGFR, all-cause mortality and ESKD (revised Table 2 and revised Figure 3). Interestingly, we observed new events (n=2) exclusively among patients in the subnephrotic group. 

Additional Editor Comments:

Both reviewers have concerned the statistical analysis.

RESPONSE: By adding p-values to Table 1 and calculating outcomes for an additional end-point (eGFR decrease by 50%), we address the points raised by both Reviewers and make up for the limited observational period.

---

## [Decision Letter · Decision Letter 1]

12 Aug 2022

The diagnostic value of native kidney biopsy in low grade, subnephrotic, and nephrotic range proteinuria: a retrospective cohort study

PONE-D-22-13404R1

Dear Dr. de Fallois 

We’re pleased to inform you that your manuscript has been judged scientifically suitable for publication and will be formally accepted for publication once it meets all outstanding technical requirements.

Kind regards,

Xianwu Cheng, M.D., Ph.D., FAHA

Academic Editor

PLOS ONE

Additional Editor Comments (optional):

All oroginal conerns have been addressed by the authors.

Reviewers' comments:

Reviewer's Responses to Questions

**Comments to the Author**

1. If the authors have adequately addressed your comments raised in a previous round of review and you feel that this manuscript is now acceptable for publication, you may indicate that here to bypass the “Comments to the Author” section, enter your conflict of interest statement in the “Confidential to Editor” section, and submit your "Accept" recommendation.

Reviewer #1: All comments have been addressed

Reviewer #2: All comments have been addressed

2. Is the manuscript technically sound, and do the data support the conclusions?

Reviewer #1: Yes

Reviewer #2: Yes

3. Has the statistical analysis been performed appropriately and rigorously? 

Reviewer #1: Yes

Reviewer #2: Yes

4. Have the authors made all data underlying the findings in their manuscript fully available?

Reviewer #1: Yes

Reviewer #2: Yes

5. Is the manuscript presented in an intelligible fashion and written in standard English?

Reviewer #1: Yes

Reviewer #2: Yes

6. Review Comments to the Author

Reviewer #1: (No Response)

Reviewer #2: (No Response)

7. PLOS authors have the option to publish the peer review history of their article (what does this mean?). If published, this will include your full peer review and any attached files.

Reviewer #1: No

Reviewer #2: **Yes: **Marios Papasotiriou MD, PhD

---

## [Editor Report · Acceptance letter]

24 Aug 2022

PONE-D-22-13404R1 

The diagnostic value of native kidney biopsy in low grade, subnephrotic, and nephrotic range proteinuria: a retrospective cohort study 

Dear Dr. de Fallois:

I'm pleased to inform you that your manuscript has been deemed suitable for publication in PLOS ONE. Congratulations! Your manuscript is now with our production department. 

Kind regards, 

on behalf of

Associate Prof. Xianwu Cheng 

Academic Editor

PLOS ONE